# Dan forms condensates in neuroblasts and regulates nuclear architecture and progenitor competence in vivo

Gillie Benchorin[1,2], Richard Jangwon Cho[2,3], Maggie Jiaqi Li[2,3], Natalia Molotkova[2,3] & Minoree Kohwi ®[2,3,4] ✉

Genome organization is thought to underlie cell type specific gene expression, yet how it is regulated in progenitors to produce cellular diversity is unknown. In *Drosophila*, a developmentally-timed genome reorganization in neural progenitors terminates competence to produce early-born neurons. These events require downregulation of Distal antenna (Dan), part of the conserved pipsqueak DNA-binding superfamily. Here we find that Dan forms liquid-like condensates with high protein mobility, and whose size and subnuclear distribution are balanced with its DNA-binding. Further, we identify a LARKS domain, a structural motif associated with condensate-forming proteins. Deleting just 13 amino acids from LARKS abrogates Dan's ability to retain the early-born neural fate gene, *hunchback*, in the neuroblast nuclear interior and maintain competence in vivo. Conversely, domain-swapping with LARKS from known phase-separating proteins rescues Dan's effects on competence. Together, we provide in vivo evidence for condensate formation and the regulation of progenitor nuclear architecture underlying neuronal diversification.

During development, progenitors are tasked with producing a multitude of cell types, each with distinct transcriptional programs. This is a particular challenge in the brain, where a small pool of neural progenitors transit through multiple states of competence to produce vastly diverse cell types in a stage-specific manner[1–5]. Higher order genome organization is thought to underlie cell type specific gene expression programs[6–10]. Yet, how genome organization contributes to stage-specific cell fate choices by the progenitor, and how it is regulated or restructured over time, are not understood. While chromatin structure itself has been the primary subjects of understanding nuclear architecture and its impacts on gene regulation, new hypotheses are emerging on the role that nucleoplasmic proteins could play in establishing or regulating genome organization through forming functional protein interaction hubs. Currently, however, major challenges in the field remain not only in identifying and

characterizing such proteins, but linking their structural properties to in vivo function.

In *Drosophila*, neuroblasts (fly neural progenitors) undergo multiple rounds of asymmetric divisions and produce a different descendent cell type with each division by expressing a series of transcription factors called temporal identity factors[11–17]. Hunchback (Hb), the first of the series, is a master regulator of early-born identity neurons, and all postmitotic early-born neurons maintain *hb* transcription as a key molecular signature of their birth order[3,7,11,18]. While *hb* is actively expressed in the neuroblast for only one to two divisions, neuroblasts remain competent to specify early-born neurons for several additional divisions. This early competence period terminates upon a mid-embryonic genome reorganization that relocates the *hb* gene to the neuroblasts' nuclear lamina for heritable silencing[3,7,12,19] (Fig. 1a), making neuroblasts particularly well-suited

[1]Department of Biological Sciences, Columbia University, New York, NY, USA. [2]Zuckerman Mind Brain Behavior Institute, Columbia University, New York, NY, USA. [3]Department of Neuroscience, Columbia University, New York, NY, USA. [4]Kavli Institute for Brain Science, Columbia University, New York, NY, USA. ✉e-mail: mk3632@columbia.edu

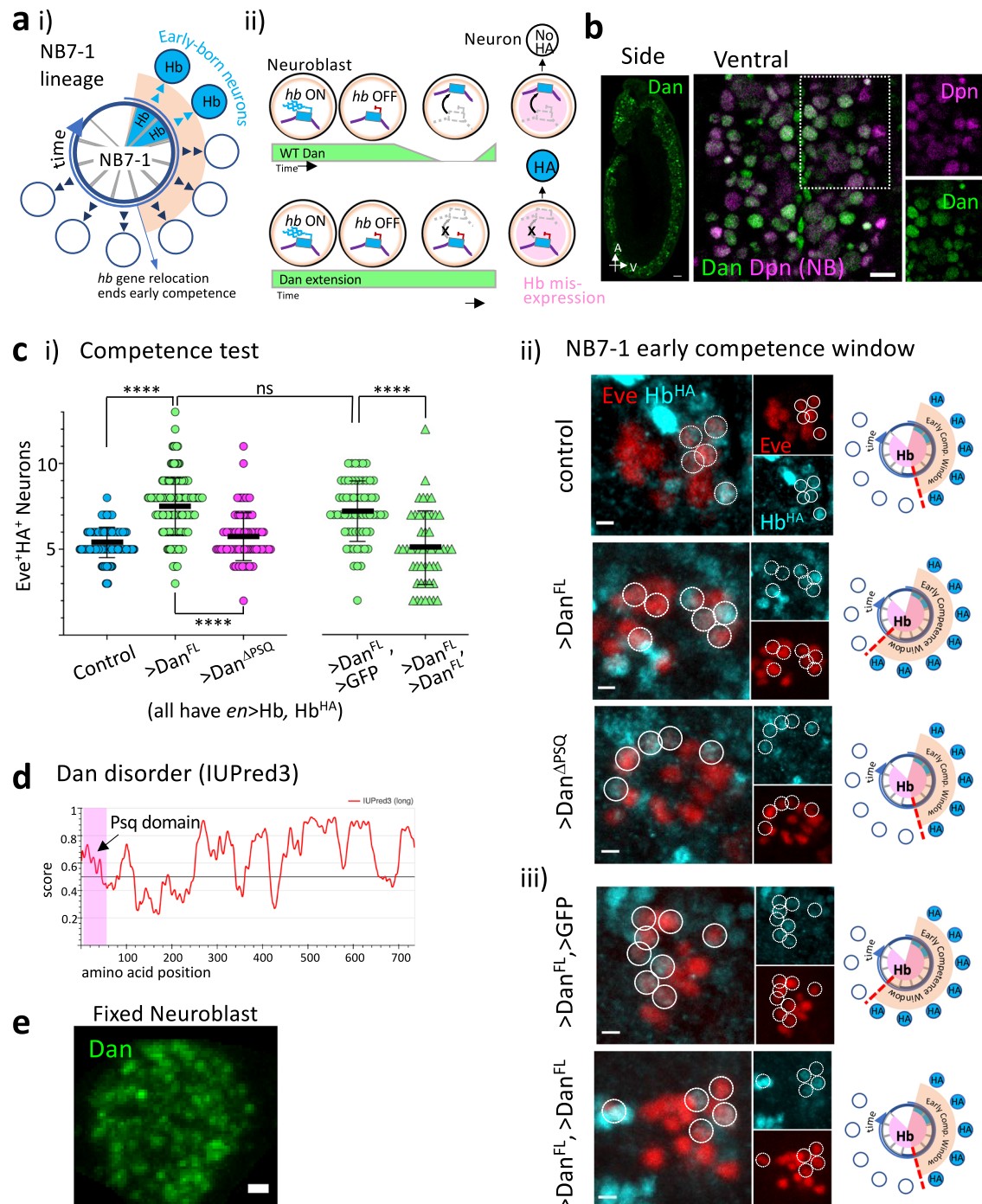

**Fig. 1 | Dan is an intrinsically disordered protein whose regulation of neuroblast competence is dose-sensitive. ai** Neuroblast 7-1 (NB7-1) and its neural lineage arranged by birth order going clockwise. The early-born neurons express Hunchback (Hb, blue). **ii** Several divisions and hours after being transcriptionally repressed, the early-born neuron gene, *hb* relocates to the nuclear lamina for heritable silencing. Prolonged expression of Distal Antenna (Dan, green) in neuroblasts blocks *hb* relocation and results in extension of early competence, as observed by their response to Hb. **b** Representative sagittal and ventral views of the embryonic ventral nerve cord at stage 11, showing Dan enrichment in the CNS and strong expression in neuroblasts (Deadpan, Dpn, pan-neuroblast marker, magenta; Dan, green). Region in dotted rectangle showing individual markers at the right. Scale bar, 10 μm. Over ten embryos have been imaged, showing Dan enrichment in the CNS. See also Kohwi et al. 2011[20]. **c** Competence test: **i** graph shows quantification of Hb^HA-expressing neural progeny of NB7-1 upon misexpression of Hb with the *engrailed*(*en*)-GAL4 driver (Eve+HA+). Error bars

represent mean ± SD (two-tailed unpaired *t* test, ****$p < 0.0001$; ns, not significant, $p = 0.2505$); control $n = 142$ NB7-1 lineages from 8 embryos, UAS-Dan^FL $n = 184$ from 10 embryos, UAS-Dan^ΔPSQ $n = 90$ from 5 embryos, UAS-GFP/UAS-Dan^FL $n = 62$ from 4 embryos, UAS-Dan^FL/UAS-Dan^FL $n = 50$ from 3 embryos; source data are provided as a Source Data file. **ii** representative images of single NB7-1 lineage shown for each genotype. Schematic diagram of early competence window results shown to the right. **iii** Dose-sensitivity of Dan on competence. Comparison of early competence window upon misexpression of one or two copies of UAS-Dan^FL. Gal4 and UAS ratios are equal between two genotypes. Quantification shown in *i*. Scale bar, 3 μm. **d** Dan intrinsic disorder prediction by IUPred3 shows that Dan is predicted to be highly disordered. The Psq domain is highlighted at the N-terminal end. **e** Dan localizes in condensates throughout the nucleus. Projection of 5 z-planes of a neuroblast immunostained for Dan in whole, fixed stage 9 embryo. Scale bar, 1μm. See also Supplementary Figs. 1, 2.

to study the role of nuclear architecture and trans-acting regulators in neurogenesis. Thus, the *hb* gene serves two purposes in the neuroblast: Initially, its transcription leads to the generation of Hb protein that acts as a cell fate determinant in the neuroblast, i.e. specify early-born neural identity. In another, parallel role, the physical position of the *hb* gene within the neuroblast nucleus determines the neuroblast's competence state, i.e., whether the descendent neuron, born two mitotic divisions later, can activate *hb* transcription[3,7]. Early competence, therefore, represents the neuroblast's internal state governed by its genome organization, which we can experimentally test for by misexpressing Hb in the neuroblast and observing whether the descendent neuron activates *hb* transcription (Fig. 1a and Supplementary Fig. 1a). Notably, misexpression of Hb directly in the postmitotic neuron does not activate *hb* transcription[3,4]; *hb*+ neurons are generated only when Hb protein is present in the neuroblast and while the *hb* gene is localized to the nuclear interior. Thus, experimentally testing for competence establishes a biological readout for the progenitor's nuclear architecture.

We previously identified a central nervous system (CNS)-specific nuclear factor, Distal antenna (Dan), that is robustly, but transiently, expressed by all neuroblasts (Fig. 1b)[20], and its downregulation at mid-embryogenesis is required for *hb* gene relocation to the neuroblast nuclear lamina and terminating early competence[3] (Fig. 1a, c). Dan is a member of a large, evolutionarily conserved superfamily of proteins characterized by a pipsqueak (Psq) helix-turn-helix motif DNA-binding domain, and Psq motif proteins have been implicated in development, chromatin regulation, and tumorigenesis[20–26]. Dan is highly enriched in neuroblasts and is most similar to CENP-B/Transposase[27], but its mechanisms of function in neuroblast nuclear architecture are unknown. Here we find that Dan is a DNA-binding protein that has high propensity to form condensates. Further, we find that Dan's ability to bind the genome and to self-associate contribute differentially to Dan protein subnuclear distribution, and are both required for Dan function in neuroblast competence. Though most of Dan protein is characterized by low complexity, disordered regions, we identified a small LARKS domain, an adhesive structural motif associated with proteins that form membraneless organelles[28,29]. Deletion of merely 13 amino acids from the LARKS domain abolished Dan's ability to retain the *hb* gene in the nuclear interior and maintain neuroblast early competence in embryos. Conversely, Dan's in vivo competence function was restored by swapping Dan's LARKS domain with that from other low complexity proteins known to undergo phase-separation. Together, these data show a nuclear factor's liquid-like self-association and its regulation of progenitor nuclear architecture underlying neural diversification.

## Results

### Dan is a low complexity protein whose regulation of neuroblast competence requires DNA-binding and is dose-sensitive

In the embryo, Dan is initially highly expressed in the ectoderm and becomes rapidly restricted to the CNS[20] (Fig. 1b). Dan is robustly, but transiently, expressed by neuroblasts, and is rapidly downregulated at mid-embryonic stage 12, concomitant with *hb* gene relocation to the nuclear lamina. Using the UAS/Gal4 system[30] to misexpress Dan in neuroblasts, thereby overriding the mid-embryonic downregulation, blocks *hb* gene relocation to the nuclear lamina and prolongs the period that neuroblasts remain competent to produce early-born neurons (Fig. 1a)[3]. Briefly, the neuroblast early competence assay is performed by quantifying the number of neurons produced with early-born identity (i.e. those that activate endogenous Hb expression) upon continuous misexpression of Hb in the neuroblast[3,7,19] (Supplementary Fig. 1a). Dan's effect on neuroblast nuclear architecture manifests in the retention of the *hb* gene in the nuclear interior, and consequently, its effects on early competence is revealed upon co-misexpression with

Hb (Fig. 1 aii and Supplementary Fig. 1a). Here, we explored mechanisms of Dan function by generating various Dan mutant constructs and performing structure-function tests on neuroblast competence. To test the effect of the various Dan mutations on early competence, we co-misexpressed the various Dan mutant constructs with Hb and compared the number of early-born neural progeny produced. We implemented phiC-mediated site-specific integration to insert all UAS transgenes into identical genomic sites. This eliminates position effects on transgene expression levels and permits direct comparison between different structure function constructs[31] (Supplementary Fig. 1b). We focused on NB7-1, one of the 30 neuroblast lineages, as this lineage has been thoroughly characterized in multiple studies in the context of temporal identity and neuroblast competence. In particular, NB7-1 is advantageous as a model lineage due to our ability to readily identify its U motoneuron progeny by their medial localization within each hemisegment of the ventral nerve cord and their expression of the Even-skipped transcription factor (Eve)[3,7,11]. Eve is expressed by a number of identifiable neurons from several neuroblast lineages, anterior/posterior corner cells (aCC/pCC) from NB1-1, RP2 neuron from NB4-2, and the U motoneurons of NB7-1, and a cluster of lateral cells from NB3-3. All are spatially separated within the nerve cord, allowing the combination of Eve expression and spatial position to identify the progeny of each lineage[11,32–34]. In the early competence assay, we used the *engrailed*-Gal4 driver to drive Hb misexpression (*en* > Hb) specifically in row 6 and 7 neuroblasts, of which only NB7-1 generates Eve+ progeny. Hb misexpression results in extra early-born neurons from multiple neuroblast lineages in which the *engrailed*-Gal4 drive is active, but supernumerary Eve+ neurons are generated only from NB7-1. These have been validated as NB7-1 progeny by their projections to larval muscle targets that are characteristic of this lineage[4,32,35,36]. All Eve+ neurons from non-NB7-1 lineages do not change in position or number, as their parental neuroblasts are not in the *engrailed* domain, and these serve as additional landmarks to identify the NB7-1 neurons in the nerve cord. Compared to the superficial and medially-positioned NB7-1 progeny, aCC/pCC/RP2 are positioned deep in the nerve cord and Eve+ neurons from NB3-3 are clustered and positioned far laterally. Thus, the combination of Eve expression and stereotyped position of the neurons unequivocally identify NB7-1 progeny.

In addition to using Eve to identify the early-born subset of the NB7-1 lineage, we detect endogenous Hb expression, to identify the early-born subset, using the Hb^HA transgenic fly line, which harbors a bacterial artificial chromosome (BAC) containing the entire *hb* gene locus that we had modified to fuse a hemagglutinin (HA) epitope tag (Supplementary Fig. 1ai). This BAC is integrated into the genome, and we have validated that both gene localization and Hb^HA expression phenocopies the endogenous *hb* gene locus[3,7]. Thus, this endogenously encoded transgene allows us to use HA as a proxy for endogenous Hb expression and distinguish it from Hb that is misexpressed from the UAS transgene[3,7,19].

Misexpression of full length Dan (Dan^FL) in neuroblasts extended competence to specify early-born identity neurons, consistent with our previous studies[3] (control misexpressing Hb alone: 5.4 Eve+HA+ neurons, $n = 8$ embryos, 142 hemisegments; co-misexpression of Hb and Dan^FL: 7.4 Eve+HA+ neurons, $n = 10$ embryos, 184 hemisegments, $p < 0.0001$) (Fig. 1ci, ii; see Supplementary Fig. 2a for genetic scheme). In contrast, misexpressing Dan in which the conserved 51 amino acid Psq sequence (Dan^ΔPSQ) has been removed, and that we found cannot bind DNA (Supplementary Fig. 1c), nearly abolished this extension (5.7 Eve+HA+ neurons, $n = 5$ embryos, 90 hemisegments, $p < 0.0001$) (Fig. 1ci, ii). Conversely, misexpressing the Psq domain alone (PSQ:GFP), was not sufficient to maintain early competence (5.3 Eve+HA+ neurons, $n = 5$ embryos, 100 hemisegments, $p = 0.2915$) (Supplementary Fig. 2b). These data indicate that Dan's Psq domain is necessary for DNA-binding, and this domain is necessary but not sufficient for Dan function in neuroblast competence.

Outside of the Psq domain, we found that Dan protein is largely characterized by low complexity, 65% as predicted by IUPred[37] (Fig. 1d). In the embryo, Dan protein forms puncta of heterogeneous size among an otherwise diffuse distribution throughout the nucleoplasm (Fig. 1e). Consistent with being intrinsically disordered, which has been demonstrated to be predictive of dosage sensitivity in yeast and animals including *Drosophila*[38], we found that Dan's ability to prolong neuroblast early competence in vivo was reduced upon misexpressing Dan from two UAS-Dan transgenes compared to one. To keep the ratio consistent between the Gal4 transcription factor and the number of UAS sequences they can bind, we compared embryos misexpressing one copy each of UAS-Dan and UAS-GFP (functionally inert control) to embryos carrying two copies of UAS-Dan. Where one copy of UAS-Dan was sufficient to extend neuroblast early competence (7.2 Eve⁺HA⁺ neurons, $n = 4$ embryos, 62 hemisegments), misexpressing two copies of UAS-Dan abolished this effect (5.1 Eve⁺HA⁺ neurons, $n = 3$ embryos, 50 hemisegments, $p = 0.1432$ compared to control) (Fig. 1ci, iii). Because the total number of UAS sites that drive protein misexpression is the same between genotypes, two copies of UAS-Dan produces more Dan protein than UAS-Dan,UAS-GFP. Thus, we conclude that Dan's function in neuroblast competence requires DNA-binding but is also dose sensitive.

## Dan subnuclear distribution is governed by both DNA-binding and self-association

To gain insights into how Dan's structural features impact its protein distribution within neuroblast nuclei, we employed microscopy in living cells. While cell fixation is a common approach for fluorescent microscopy, recent studies have shown that fixation can potentially introduce artifacts that alter the way proteins appear to localize or behave. This issue may be particularly concerning for proteins with intrinsically disordered regions that form multivalent and transient interactions[39,40]. Thus, to avoid potentially introducing fixation artifacts, we took advantage of S2 cells, a *Drosophila* embryo-derived cell line, to perform live-imaging of Dan protein by expressing GFP-fused Dan constructs (Dan:GFP). Notably, S2 cells do not express Dan endogenously[41], and thus Dan:GFP distribution would not be influenced by an existing Dan protein population. We first expressed GFP alone and confirmed that GFP distributes uniformly throughout the nucleus and does not form any visible puncta or aggregates (Fig. 2ai). In contrast, full length Dan (Dan^FL:GFP) in live S2 cells formed discrete puncta among a more diffuse distribution, highly similar to endogenous Dan distribution in fixed neuroblasts in embryos (Fig. 2ai vs. Fig. 1e). We functionally tested Dan:GFP using the early competence assay in embryos, and we found that misexpression of Dan:GFP results in similar extension of early competence as Dan that is not fused to any tag (Fig. 2aiii). These observations support that Dan:GFP formation of discrete puncta is due to Dan protein, and the GFP tag is functionally inert and does not interfere with or alter Dan function in neuroblasts.

We next generated several mutant Dan constructs to express in live S2 cells. All constructs were derived from the same GFP-fusion backbone to ensure that any differences in Dan:GFP distribution are the result of changes to the Dan protein. As described above, Dan^FL formed small puncta among a diffusely distributed protein in nuclei, and consistent with Dan being a DNA-binding protein, both of these populations co-localized with chromatin in S2 cells, just as endogenous Dan protein in neuroblasts (Supplementary Fig. 3). We found that the diffuse distribution was completely lost upon deletion of the Psq domain (Fig. 2aii, PSQ domain is highlighted in pink), and Dan protein coalesced into several large hubs (Dan^ΔPSQ:GFP, Fig. 2ai). Consistent with the lack of a DNA-binding domain (DBD), Dan^ΔPSQ:GFP hubs avoided chromatin (Supplementary Fig. 3). To further analyze the relationship between DNA-binding and condensate formation, we next examined a more subtle mutation in the DBD. The residues at the beginning of the

third helix confer the core DNA-binding function[42], and a glutamic acid-to-lysine point mutation in the DBD (Dan^E45K, Fig. 2aii, blue box) has been shown previously to reduce but not abolish function of Psq family proteins, producing a hypomorphic *dan* allele[23,43,44]. We found that the Dan^E45K mutation resulted in a similar, but weaker, phenotype, in which there was a decrease in the diffuse distribution and an increase in size of Dan puncta compared to full length Dan (Fig. 2ai, Dan^E45K:GFP). Dan^E45K:GFP, and especially Dan^ΔPSΔPSQ:GFP condensates, which are larger than the diffraction limit, were highly circular (Fig. 2bi, left plot)[45–47]. By comparing cells expressing similar levels of Dan protein based on similar nuclear GFP signal, we found that decreased DNA-binding increased condensate size while reducing their number (Fig. 2bi, middle and right plots). Further, condensate size increased with higher protein concentration (Fig. 2bii). In addition to protein coalescence, the large Dan^ΔPSQ:GFP hubs were found preferentially localized at the nuclear periphery. While Dan^FL:GFP foci are distributed throughout the nucleus relatively uniformly ($n = 3$ cells, 15 measurements of pixel intensity through a 3D projection of the cells), the average peak pixel intensity of Dan^ΔPSQ:GFP hubs ($n = 3$ cells, 20 hubs measured) predominantly lay near the nuclear edge (Fig. 2c). Thus, Dan protein has strong propensity to self-associate, and the weaker its interaction with DNA, the more the self-association dominates its subnuclear distribution. When completely released from DNA-binding, Dan coalesces into large hubs that localize at the nuclear periphery. We conclude that Dan's DNA-binding and self-association properties contribute differently to its protein distribution within nuclei.

## Dan has liquid-like properties and is characterized by high protein mobility

To further test protein mobility, we performed fluorescence recovery after photobleaching (FRAP). Highly mobile proteins can recover fluorescence on the scale of seconds to minutes, while solid aggregates can take tens of minutes to hours to turn over[48,49]. We found that Dan^ΔPSQ:GFP condensates undergo signal recovery on the order of seconds (Fig. 3a), indicating a rapid exchange of protein[49–51]. This was in stark contrast to Histone2B:GFP (H2B:GFP), a stably bound protein that is commonly used as a FRAP control, which does not recover fluorescence on the time scale tested (Fig. 3a, Supplementary Fig. 4aii, and Supplementary Videos 1 and 2)[52–55]. Dan^FL:GFP signal also showed rapid signal recovery after photobleaching (Fig. 3a plot and Supplementary Fig. 4a), indicating that Dan protein high mobility is not a characteristic specific to a non-DNA bound state. Furthermore, we found that signal recovery of the small condensates (Dan^E45K:GFP, Supplementary Fig. 4a and Supplementary Video 3) was faster than large condensates (Dan^ΔPSQ:GFP). This is consistent with proteins in which diffusion dominates protein interactions and is a measure of protein liquidity[48], as larger volume protein clusters require more diffusion and exchange between bleached and unbleached protein than smaller clusters[51]. Additionally, we found that Dan^ΔPSQ condensates can fuse when in close contact (Fig. 3b and Supplementary Video 4). Together, these properties support a liquid-like nature of Dan protein condensates.

We found that repeated photobleaching, or FLIP (fluorescence loss in photobleaching)[56–58], of a single Dan^ΔPSQ:GFP condensate led to a gradual loss of fluorescence in all of the condensates in the cell (Fig. 3c). We sequentially photobleached a single condensate four times and imaged the whole nucleus three-dimensionally between each bleaching cycle. The same settings (area, laser intensity, and duration) were used for each bleaching event in both experimental and control cells. The total fluorescence intensity in control cells, which were bleached just outside the nucleus, remained stable over time, or even slightly increased over the four bleach cycles. In contrast, repeated photobleaching of a single, targeted condensate resulted in a gradual and significant reduction in fluorescence intensity of all the condensates, and a decrease in total nuclear fluorescence over time

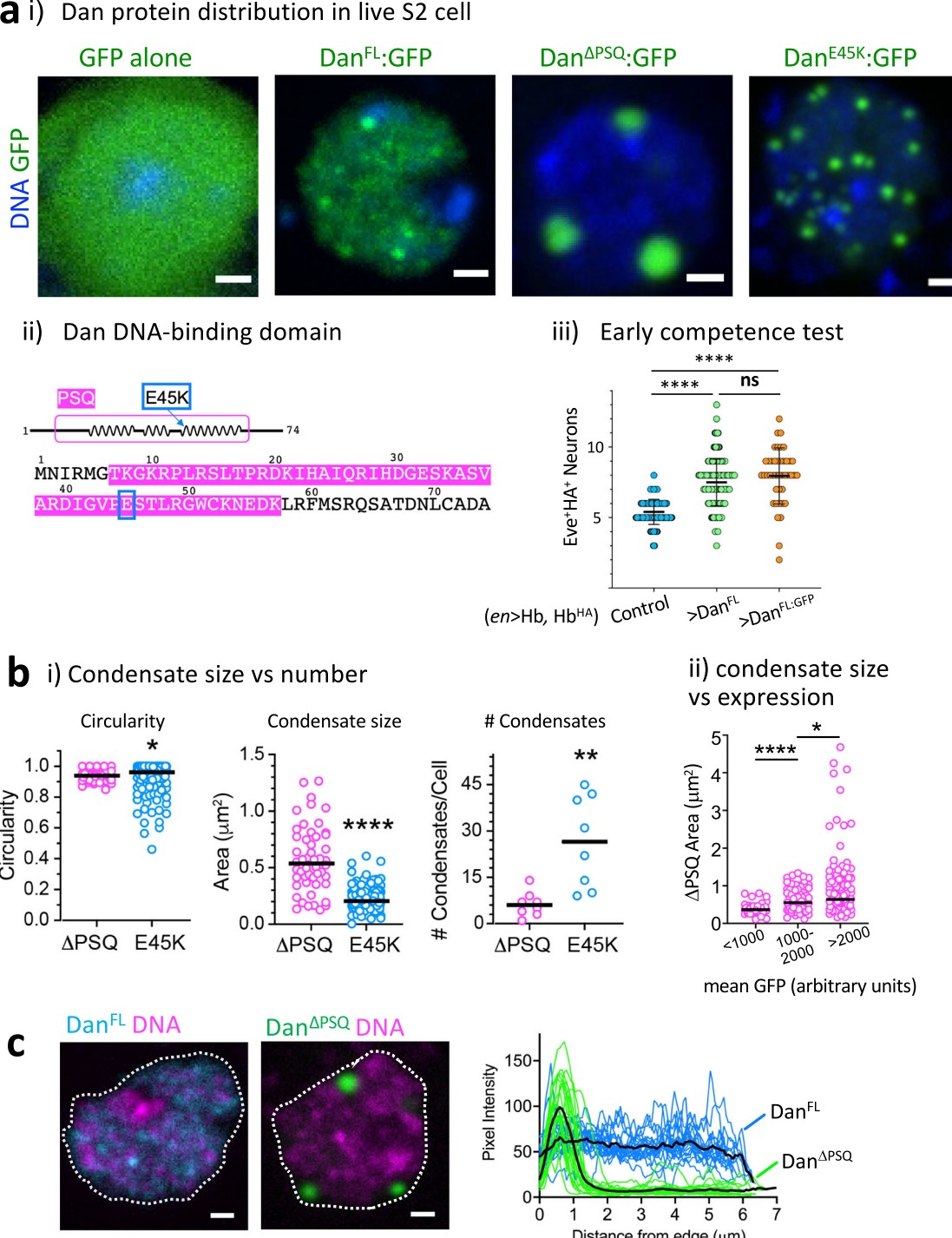

**Fig. 2 | DNA-binding and self-association properties differentially contribute to Dan subnuclear distribution. ai** Dan protein distribution in live S2 cells. From left: GFP alone, full length Dan fused to GFP (Dan$^{FL}$:GFP), Dan$^{\Delta PSQ}$:GFP, and Dan$^{E45K}$:GFP. DNA marked with NucBlue (blue). Scale bars, 1 μm. **ii** Schematic of the N-terminus region of Dan, showing the conserved Psq DNA-binding domain, including the three helices (magenta). The E45K mutation occurs at the beginning of the third alpha helix (blue box). **iii** Early competence test showing that Dan:GFP (orange) has same effects on competence as Dan with no GFP tag (green). Graph shows quantification of Hb$^{HA}$-expressing neural progeny of NB7-1 (Eve$^+$HA$^+$) upon misexpression of Hb. Error bars represent mean ± SD (two-tailed unpaired *t* test, ****$p < 0.0001$; ns, not significant, $p = 0.1213$); control $n = 142$ NB7-1 lineages from 8 embryos, UAS-Dan$^{FL}$ $n = 184$ from 10 embryos, UAS-Dan$^{GFP}$ $n = 42$ from 3 embryos; source data are provided as a Source Data file. **bi** Graphs comparing Dan$^{\Delta PSQ}$:GFP and Dan$^{E45K}$:GFP expressing S2 cells with comparable GFP expression levels. Left plot shows condensate circularity (two-tailed Mann–Whitney, *$p = 0.0404$), and

middle plot shows condensate size (two-tailed Mann–Whitney, ****$p < 0.0001$); ΔPSQ $n = 50$, E45K $n = 170$ condensates. Right plot shows number of condensates per cell among $n = 8$ cells tested per condition (two-tailed unpaired *t* test, **$p = 0.0056$). Black bar represents median. **ii** Plot showing increasing condensate size (Dan$^{\Delta PSQ}$:GFP) with increasing protein concentration (two-tailed Mann–Whitney, ****$p < 0.0001$, *$p = 0.0472$). Black bar represents the median; <1000 $n = 40$ condensates from four cells, 1000–2000 $n = 69$, >2000 $n = 106$. Source data are provided as a Source Data file. **c** Representative image of S2 cells expressing Dan$^{FL}$:GFP (blue) or Dan$^{\Delta PSQ}$:GFP (green). The large Dan$^{\Delta PSQ}$ condensates preferentially localize at the nuclear periphery. DNA marked with NucBlue (pink). Dotted line outline nuclear edge. Scale bars, 1μm. Graph shows GFP signal intensity plotted as distance from edge of nucleus. 20 Dan$^{\Delta PSQ}$ (green lines) and 15 Dan$^{FL}$ (blue lines) condensates were traced from three cells each. Black line shows the mean of the traces for each. Source data are provided as a Source Data file. See also Supplementary Fig. 3.

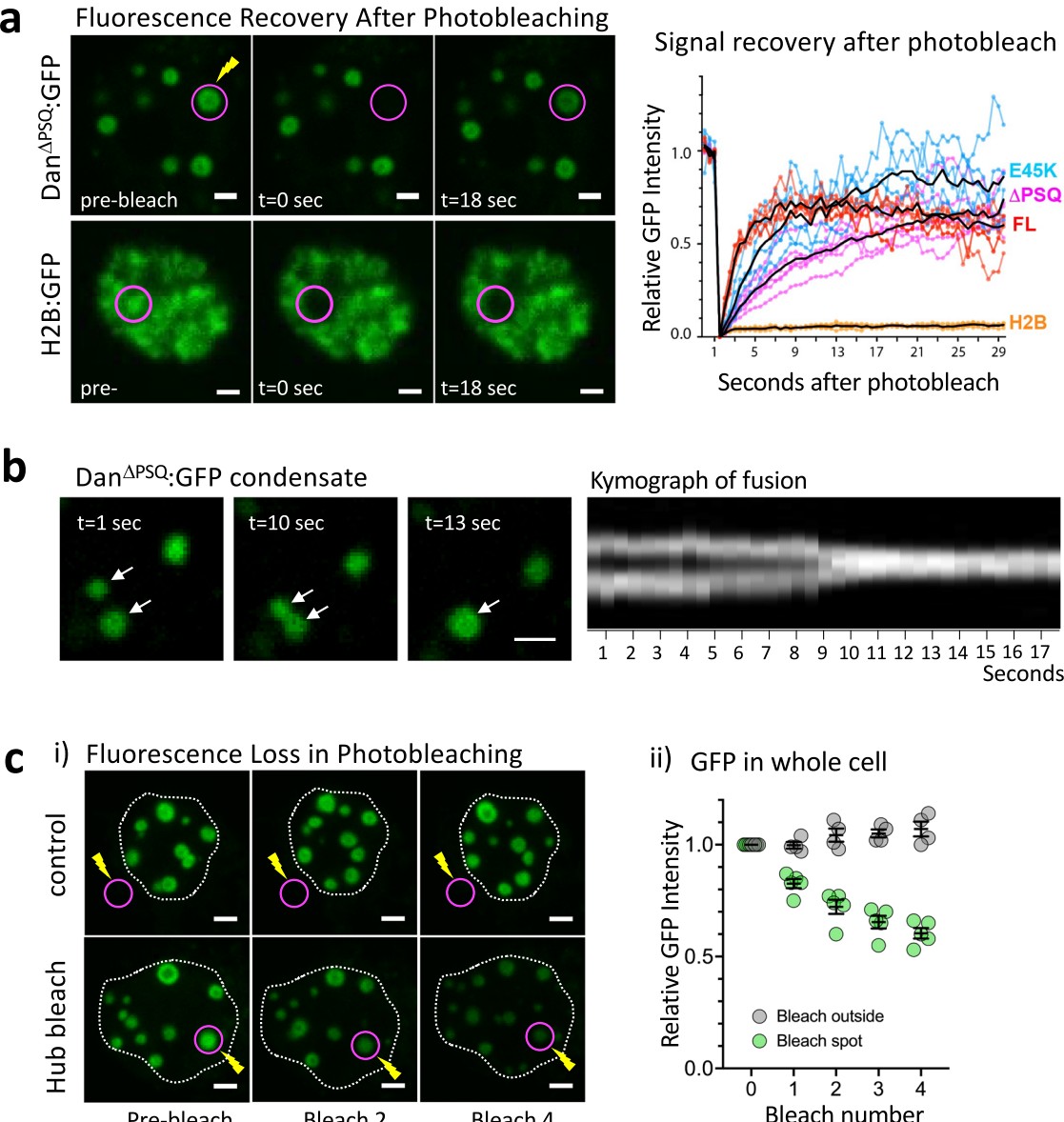

**Fig. 3 | Dan protein shows liquid-like properties and fast protein mobility.**
**a** Representative images of Fluorescence recovery after photobleaching (FRAP) in
S2 cells are shown before, during, and after photobleaching a Dan$^{\Delta PSQ}$:GFP con-
densate or H2B:GFP region (see supplemental material for video files). Graph shows
quantification of signal recovery of Dan$^{FL}$:GFP ($n = 6$ condensates), Dan$^{\Delta PSQ}$:GFP
($n = 6$), Dan$^{E45K}$:GFP ($n = 5$), and H2B:GFP control ($n = 4$). Black line shows the mean
of the traces for each. Source data are provided as a Source Data file. **b** Images from
a video showing two Dan$^{\Delta PSQ}$:GFP droplets before, during, and after fusion. The
kymograph shows the relative position of the droplets over time. **ci** Representative

z-projection images of Dan$^{\Delta PSQ}$:GFP in S2 cells before and after FLIP (bleaching
repeated four times), targeting either a single droplet or a single point just outside
the nucleus, as indicated by the magenta circle. **ii** Quantification of total relative
fluorescent signal from a z-projection through the entire S2 cell after FLIP. Starting
GFP = 1 at time 0 corresponds average of the first three images before bleaching.
$n = 4$ cells were imaged per condition. Error bars represent mean ± SD; source data
are provided as a Source Data file. All scale bars, 1 μm. See also Supplemen-
tary Fig. 4.

(Fig. 3cii). Notably, each bleach event completely abolished GFP signal
from only the target condensate and had no immediate effect on
neighboring condensates within a five second period, but after
30 seconds, the fluorescent intensity of neighboring condensates
decreased (Supplementary Fig. 4b). Thus, while Dan$^{\Delta PSQ}$:GFP con-
densates appear as independent units, they are highly interconnected,
with individual Dan protein molecules rapidly moving between them.

**A LARKS domain is essential for Dan's ability to regulate com-
petence in vivo**
Because Dan is a largely disordered protein (Fig. 1d), it is likely that
most of the protein besides the DBD contributes in some capacity to its
self-association propensity. Indeed, the Psq domain alone, which

effectively removes all of the disordered regions, is relatively uni-
formly distributed within S2 cell nuclei compared to full length Dan
(Fig. 4ai, PSQ:GFP). Within the expansive disordered regions of the
protein, however, we found a small, predicted LARKS domain, a low-
complexity structural motif characterized by weak self-interactions.
LARKS domains form reversible interactions through kinked beta-
sheets and have been attributed to dynamic protein organization and
functional aggregation[28,29,59]. Dan's LARKS domain is defined by a 13
amino acid stretch of primarily glycines, just downstream of the Dan
Psq motif (Fig. 4aii). Given that the LARKS domain constitutes only a
small fraction of the predominantly disordered protein, and the DBD
is a strong component of Dan protein localization, we would not expect
LARKS deletion to visibly alter Dan protein distribution at the

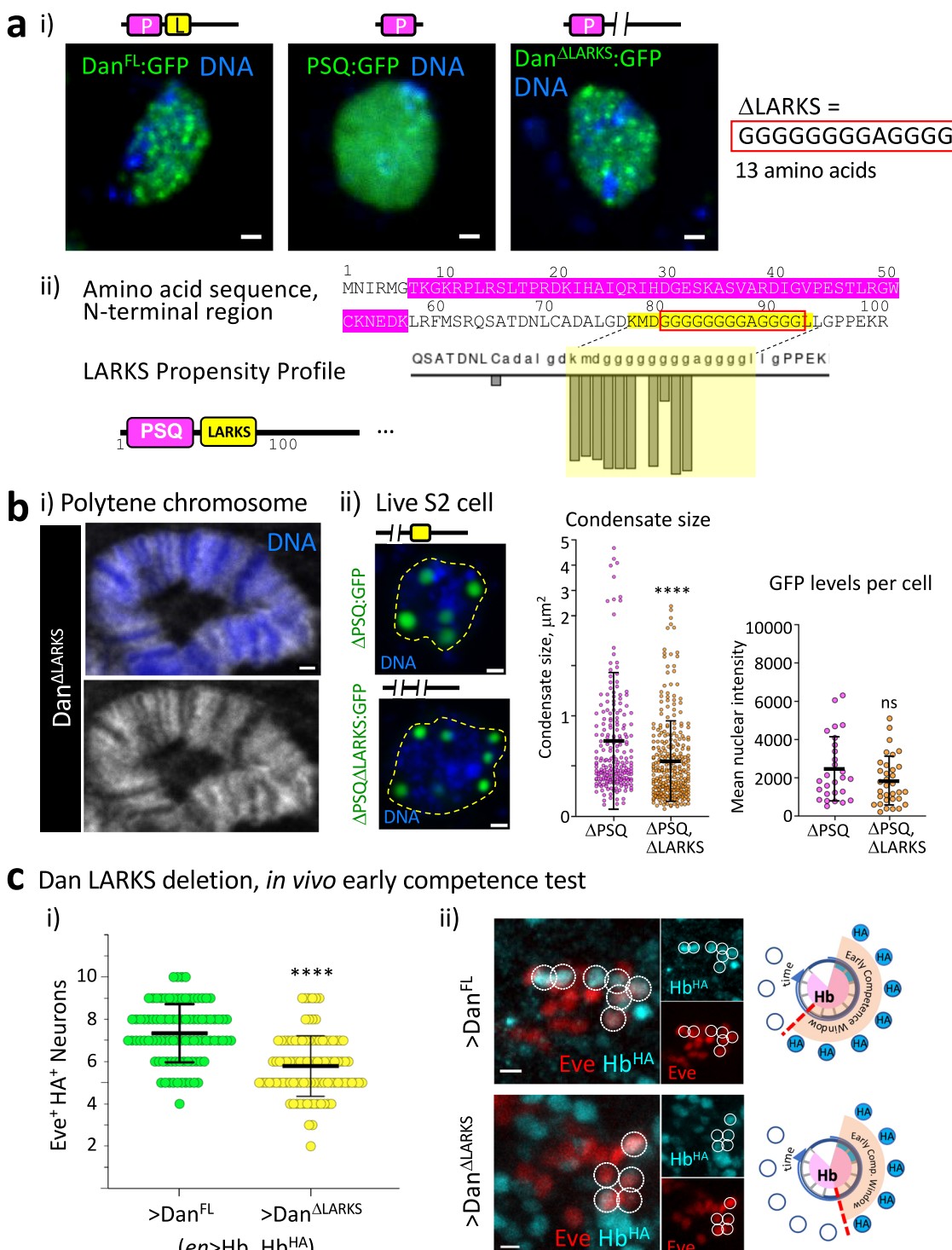

**Fig. 4 | Dan has a LARKS domain that is essential for its ability to regulate competence in vivo. ai** Live-imaged S2 cells expressing Dan$^{FL}$:GFP, PSQ:GFP, or Dan$^{\Delta LARKS}$:GFP. Construct schematic is shown above each image. **ii** Schematic diagram showing the N-terminal region of Dan, indicating the PSQ domain (magenta) and the LARKS domain (yellow). The amino acids in the red box indicate those that are deleted in ΔLARKS. The LARKS propensity profile (see methods section for link) shows the predicted glycine-rich LARKS domain (yellow highlight) **bi** Portion of a polytene chromosome image stained with myc (Dan$^{\Delta LARKS}$) and DAPI (DNA stain) showing that Dan still binds the genome without the LARKS domain. Scale bar, 1 μm. **ii** Example cells showing Dan$^{\Delta PSQ}$:GFP and Dan$^{\Delta PSQ,\Delta LARKS}$:GFP condensates;

yellow dotted line shows the nuclear border. Scale bars, 1 μm. Graph shows quantification of Dan$^{\Delta PSQ}$ and Dan$^{\Delta PSQ,\Delta LARKS}$ condensate sizes among cells expressing comparable levels of GFP; condensates: ΔPSQ $n = 215$ and ΔPSQ,ΔLARKS $n = 301$; cells: ΔPSQ $n = 27$ and ΔPSQ,ΔLARKS $n = 33$; Error bars show mean ± SD; two-tailed Mann−Whitney, ****$p < 0.0001$, ns $p = 0.1825$. **ci** Quantification of early competence window comparing effects of Dan$^{FL}$ ($n = 94$ NB7-1 lineages from 5 embryos) to Dan$^{\Delta LARKS}$ ($n = 100$ from 6 embryos). Error bars represent mean ± SD; two-tailed unpaired $t$ test, ****$p < 0.0001$. Source data are provided as a Source Data file. **ii** Images of single representative NB7-1 lineage early-born neurons (Eve$^+$HA$^+$), Scale bars, 3 μm.

macroscopic level. Indeed, Dan[ΔLARKS]:GFP still showed discrete puncta among a more diffuse distribution, similar to Dan[FL]:GFP (Fig. 4ai). Further, we expressed Dan[ΔLARKS] in salivary gland cell polytene chromosomes, and we confirmed that Dan[ΔLARKS] still binds the genome (Fig. 4bi), in contrast to Dan[ΔPSQ] (Supplementary Fig. 1c), indicating that loss of the LARKS domain does not interfere with Dan interaction with the genome. To allow for a more nuanced measurement of the effects of LARKS deletion on Dan self-association properties, we deleted the 13 amino acids from the LARKS domain from the Dan[ΔPSQ] construct, which in S2 cells displays little diffuse distribution, and the hubs are large and well-defined (resulting in a Dan[ΔPSQ,ΔLARKS]:GFP construct). We found that between groups of cells expressing comparable levels of Dan protein, there was a small but measurable decrease in condensate size (Fig. 4bii), consistent with what is known about LARKS domain properties to facilitate protein interactions. Thus, we conclude that without the LARKS domain, Dan protein can still bind DNA, but its ability to self-associate is reduced. In the presence of DNA-binding activity, which perhaps may be the primary driver of Dan distribution within nuclei, LARKS deletion does not result in an broad loss of coalescence that is readily visible at the macroscopic level of imaging the whole nucleus. However, LARKS deletion may affect Dan's ability to efficiently form hubs at the smaller, genomic scale. Indeed, while condensates are often studied in vitro as visibly large protein aggregates, it may be that critical functions might occur at scales that we cannot observe by microscopy[60]. In order to determine whether the LARKS domain is required for Dan function in vivo, we implemented the early competence assay in embryos. Dan[FL] can extend neuroblast early competence by retaining the *hb* gene in the nuclear interior. We found that deleting the LARKS domain (Dan[ΔLARKS]) was sufficient to abolish Dan's effect on competence (Dan[FL]: 7.3 Eve+HA+ neurons, $n = 5$ embryos, 94 hemisegments; Dan[ΔLARKS]: 5.8 Eve+HA+ neurons, $n = 6$ embryos, 100 hemisegments, $p < 0.0001$) (Fig. 4c). Thus, the glycine-rich 13 amino acids of the LARKS domain is essential to mediate Dan's competence function in embryos.

### LARKS is essential for Dan's ability to regulate radial gene relocation in neuroblasts

Given the loss of Dan's competence phenotype upon deletion of the LARKS domain, we examined *hb* gene radial relocation by immuno-DNA Fluorescence in situ Hybridization (FISH) on intact embryos. Briefly, we generated a fluorescently-tagged DNA FISH probe for the *hb* gene and hybridized control (yw) embryos and embryos misexpressing either Dan[FL] or Dan[ΔLARKS], We analyzed embryos at stage 12 mid-embryogenesis, when the *hb* gene normally becomes relocated to the nuclear periphery[3]. In addition to hybridizing with the DNA probe for the *hb* gene, we also immunostained these embryos with lamin Dm0 (*Drosophila* Lamin B) to detect the nuclear envelope, and Worniu, a pan-neuroblast marker. These allowed us to detect the subnuclear position of the *hb* gene relative to the nuclear periphery specifically in the neuroblast population of intact embryos. Embryonic stages were identified based on their stereotyped and well-defined morphologies[61]. We found that compared to control neuroblasts, the *hb* gene was retained in the nuclear interior upon expression of Dan[FL], but not Dan[ΔLARKS] (YW: $n = 4$ embryos/213 *hb* loci; UAS- Dan[FL]: $n = 5$ embryos/ 275 *hb* loci; UAS- Dan[ΔLARKS]: $n = 3$ embryos/191 *hb* loci). The fraction of *hb* gene loci either at or near the neuroblast nuclear lamina was similar between control embryos and embryos misexpressing Dan[ΔLARKS] (Fig. 5a), consistent with their early competence phenotypes. Thus, we conclude that the LARKS domain is essential for Dan to regulate neuroblast competence through *hb* gene relocation.

LARKS domains are protein structures common among low complexity domains of proteins that are found in membraneless organelles and mediate reversible interactions, a cornerstone of proteins that undergo liquid-liquid phase-separation (LLPS). To confirm whether Dan's LARKS domain functions through self-

association, we performed the neuroblast competence assay misexpressing Dan in which its LARKS domain was swapped for other protein aggregation domains. In the first, we replaced the glycine-rich LARKS domain with the poly-asparagine repeat (Dan[NNN]), which is found in high abundance in prions and other proteins that also aggregate[62–65]. Consistent with such aggregation properties, in S2 cells the asparagine stretch was able to increase Dan protein coalescence (Supplementary Fig. 5). Importantly, we found that the LARKS-to-NNN domain swap was sufficient to completely rescue Dan's ability to extend competence in vivo (Dan[NNN]: 7.2 Eve+HA+ neurons, $n = 89$ NB7-1 lineages from 5 embryos, $p < 0.0001$) (Fig. 5b). We performed two additional domain-swap experiments with LARKS domains from other proteins, FUS[28], a well-studied protein known to undergo LLPS, and the nuclear pore complex protein Nup98, which contains an FG-repeat necessary for LLPS[29,66]. Both of these domains also rescued Dan's competence phenotype in embryos (Dan[FUS] swap: 8.8 Eve+HA+ neurons, $n = 45$ from 3 embryos, $p < 0.0001$; Dan[Nup98] swap: 7.6 Eve+HA+ neurons, $n = 65$ from 4 embryos, $p < 0.0001$) (Fig. 5b). In all cases, the Dan LARKS domain was replaced by a donor LARKS region of similar size. Thus, we conclude that the Dan LARKS domain is an essential component of Dan, necessary for Dan's ability to regulate *hb* gene relocation to the neuroblast nuclear lamina and maintenance of the early competence state in the embryo. Our data suggests that this function is mediated by LARKS promoting Dan self-association. In sum, Dan's DNA-binding and self-association contribute differentially to Dan subnuclear distribution, and Dan condensate formation properties regulate neuroblast nuclear architecture underlying neural diversification (summary and model in Fig. 6).

## Discussion

The three-dimensional organization of the genome is thought to underlie cell type specific gene expression, and changes in this organization can impact cell function or identity. Nuclear architecture and its role in gene regulation has largely focused on proteins of the nuclear envelope and chromatin, but how nucleoplasmic proteins might function to allow stage-specific reorganization is unknown, and there is a crucial lack of in vivo studies that demonstrate the role of such mechanisms in biological processes. Here we have taken a reductionist approach to address these questions in vivo by focusing on a neuroblast nuclear factor Dan, and its regulation of the *hb* gene, a master regulator gene of early-born neurons. In embryos, the developmentally-timed radial re-positioning of the *hb* gene to the neuroblast nuclear periphery is the critical determinant to close the neuroblast's period of competence to produce early-born identity neurons. Further, *hb* gene relocation requires the downregulation of the neuroblast nuclear factor Dan[3,20]. Here we found that Dan protein has liquid-like condensate properties and find a domain that is essential for regulating neuroblast nuclear architecture and competence in vivo.

Dan is a Psq motif DNA-binding protein that displays widespread genome binding, as we observed in our polytene chromosome spread stains. In cells, Dan normally shows a diffuse distribution with puncta of varying sizes, and upon impairment of the DBD, coalesces into larger condensates. Computer simulations have recently shown how nucleoli-chromatin interaction can impact nucleoli coalescence and condensate size. The authors' model show that nucleoli-chromatin interactions can facilitate condensate nucleation, but chromatin network constraints create an energetic barrier that limit coalescence[67]. A recent study examined the intrinsically disordered regions (IDRs) of Polyhomeotic, a polycomb group (PcG) complex PRC1 subunit, and found that deleting a central, glutamine-rich IDR resulted in large condensates that excluded chromatin[68], similar to what we observed with Dan[ΔPSQ], suggesting that chromatin interaction and condensation are in balance. Interestingly, while Dan coalesces into a few large hubs when the entire DBD is removed, a point mutation in the DBD that

## a  DNA FISH

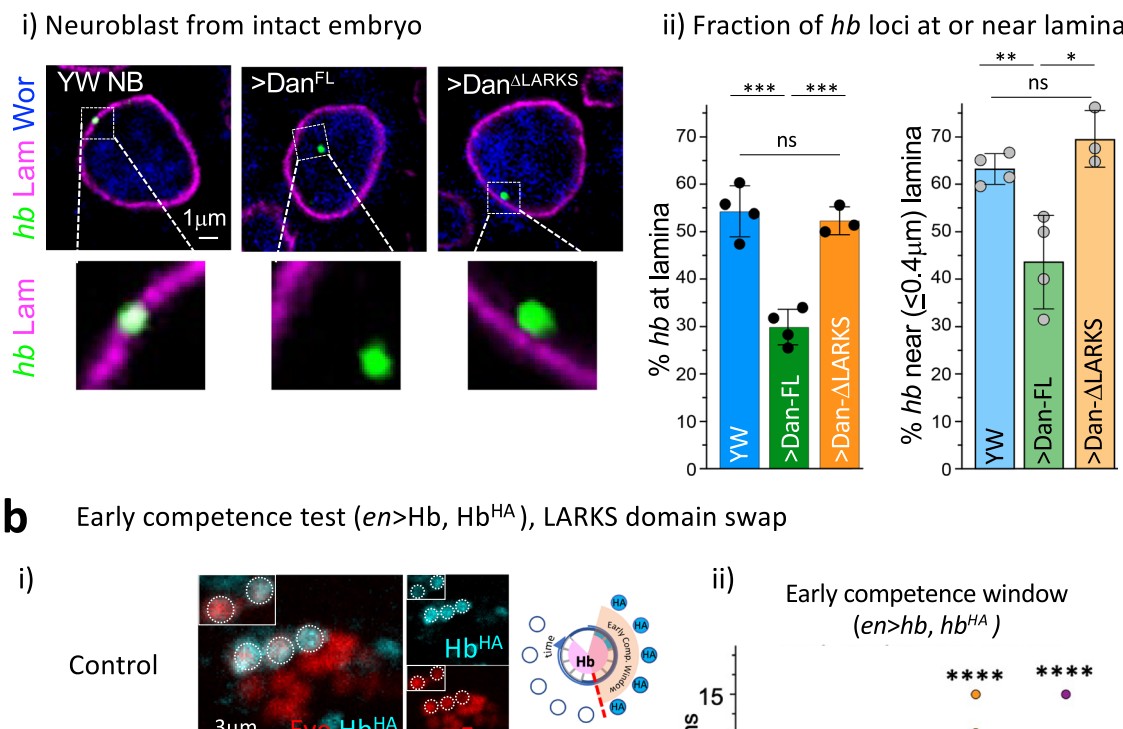

### i) Neuroblast from intact embryo

### ii) Fraction of *hb* loci at or near lamina

## b  Early competence test (*en*>Hb, Hb^HA), LARKS domain swap

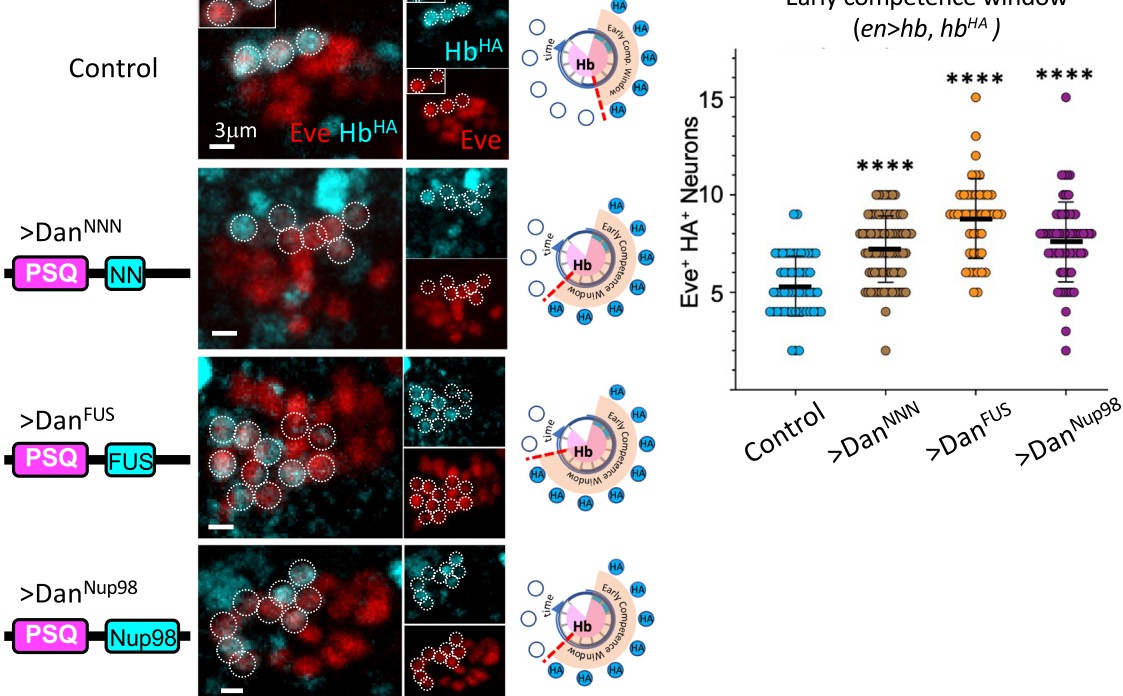

**Fig. 5 | Dan's LARKS domain is essential for Dan's ability to regulate radial gene relocation in neuroblasts. ai** Representative DNA FISH images of a single z-plane through a neuroblast nucleus within an intact stage 12 embryo, showing the *hb* gene locus (green), nuclear lamin (lamin Dm0, magenta), and the pan-neuroblast marker, Worniu (Wor, blue). Embryos misexpressing Dan^FL or Dan^ΔLARKS with the pan-neuroblast GAL4 driver, *insc*-GAL4, are compared to wild type (YW). Magnified images (*hb* FISH and lamin) of the region in the white box are shown below each image. **ii** Graphs show quantification of the DNA FISH images shown in *i*. Left graph shows fraction of *hb* loci at nuclear lamina (pixel overlap between *hb* FISH signal and lamin signal), and right graph shows fraction of *hb* loci near ( ≤0.4 μm) lamina; YW *n* = 4 embryos/213 *hb* loci, Dan^FL *n* = 5 embryos/275 *hb* loci, Dan^ΔLARKS *n* = 3

embryos/191 *hb* loci. Error bars represent mean ± SD. Two-tailed unpaired *t* test: on-lamin, YW-Dan^FL ***p = 0.0003, Dan^FL-Dan^ΔLARKS ***p = 0.0004, YW-Dan^ΔLARKS ns p = 0.5906; near-lamin, YW-Dan^FL **p = 0.0094, Dan^FL-Dan^ΔLARKS *p = 0.0155, YW-Dan^ΔLARKS ns (p = 0.1654). Source data are provided as a Source Data file. **bi** Images of single representative NB7-1 lineage early-born neurons (Eve^+HA^+) from embryos misexpressing various Dan constructs with the LARKS domain swapped. Scale bar = 3 μm. **ii** Graph shows quantification of early competence window. Each data point is a single NB7-1 lineage. Error bars represent mean ± SD. Two-tailed unpaired *t* test: YW *n* = 54 from 3 embryos, Dan^NNN *n* = 89 from 5 embryos, Dan^FUS *n* = 45 from 3 embryos, Dan^Nup98 *n* = 65 from 4 embryos; ****p < 0.0001. Source data are provided as a Source Data file. See also Supplementary Fig. 5.

causes a weak, hypomorphic allele adopts an intermediate distribution, with a reduction in the diffuse distribution and an increase in puncta sizes compared to full length Dan. Thus, it appears that Dan protein is pulled towards genome binding and self-association simultaneously, rather than existing as separate genome-binding

versus coalescing protein subsets. Consistent with this, our FRAP data of Dan^FL and Dan^ΔPSQ showed similarly fast kinetics of signal recovery. Perhaps the heterogeneous condensate distribution of Dan within nuclei reflects differences in local chromatin structure that impose constraints on Dan coalescence. The experiments

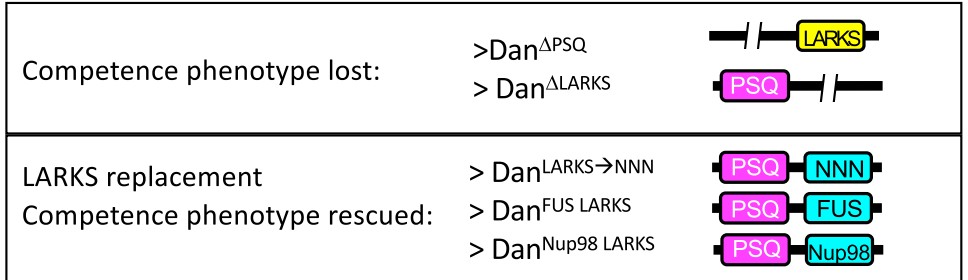

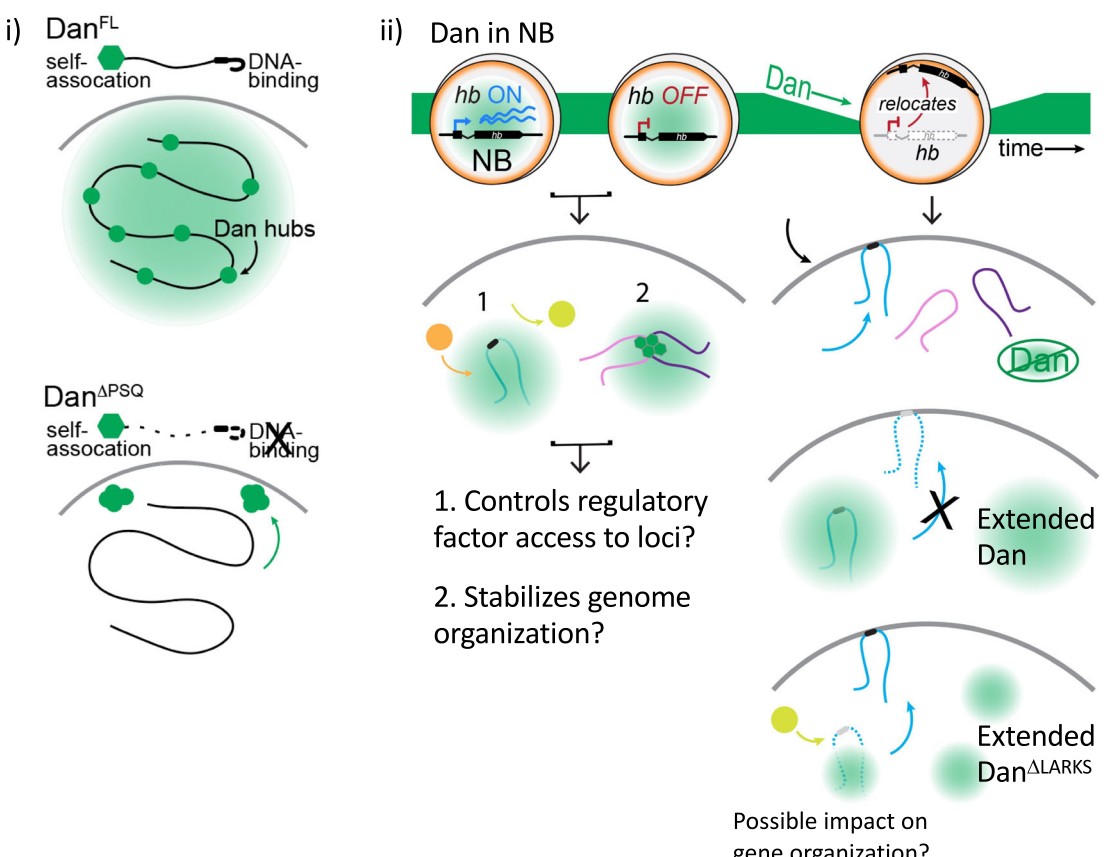

**Fig. 6 | Summary and model. a** Summary of Dan constructs and their effects on neuroblast competence. **b** Schematic model of results. **i** Dan is a protein that binds widely in the genome and also forms small condensates. Upon release from DNA-binding, Dan coalesces into large condensates that preferentially localize at the nuclear periphery. **ii** In the embryo, neuroblasts express high levels of Dan initially, and rapidly downregulate Dan at mid-embryogenesis allowing gene reorganization.

While LARKS deletion partially affects Dan coalescence, it abrogates Dan's ability to retain the *hb* gene in the nuclear interior and maintain the early competence state. This may occur through reduced efficiency to tether some target genes to Dan hubs in the nuclear interior and/or to restrict their interaction with other gene regulators. Future studies are required to determine mechanisms underlying Dan regulation of genome organization.

presented in this study lay the groundwork for understanding Dan protein properties and give insights into the apparent balance that exists between Dan propensity to interact with DNA and its propensity to interact with other Dan proteins. As future follow-up studies, it would be important to introduce these structural mutations in the endogenous *dan* gene. This would allow us to study how the different Dan protein domains impact its behavior and nuclear distribution at endogenous expression levels and within its normal cellular environment of the neuroblast, which may play a role in regulating Dan protein accumulation and balance between genome-binding and coalescence.

Whether protein assemblies form as a consequence of phase-separation or other modes of protein clustering is a topic of controversy[48,69]. Overall, we found Dan condensates appear to have liquid-like properties, as they are highly circular, have fast FRAP signal recovery rates, and can fuse, though further studies will be needed to differentiate between truly LLPS properties versus specific protein interactions with fast kinetics. Whatever the physical mechanisms underlying protein interactions, there is still a great need for understanding whether and how self-association properties of nuclear proteins contribute to genome organization and how it impacts the transcriptional state of cells in a biological context. Dan displays a

temporally dynamic expression pattern in neuroblasts, initially being highly expressed across the neuroblast population and then becoming rapidly downregulated at mid-embryogenesis, concomitant with *hb* gene relocation to the nuclear lamina. Misexpressing Dan to override this downregulation maintains the neuroblast in the early competence state. These features provide a unique opportunity to study protein coalescence in nuclear architecture in vivo. Interestingly, we found that misexpressing higher levels of Dan, using two UAS transgenes instead of one, reversed this competence extension phenotype. Work by Vavouri et al. showed in *Drosophila* and C. *elegans* that disordered proteins make promiscuous interactions upon an increase in concentration and is likely the underlying cause of pathology when genes are overexpressed[38]. Perhaps Dan's dose-sensitivity arises from a similar explanation, in which higher levels of Dan could sequester Dan protein away from binding its genomic targets or sequester other yet unidentified molecular factors from acting on the *hb* locus for radial relocation. These are interesting questions to address in the future.

While Dan is characterized by predominantly low-complexity regions, we found near the DBD a small LARKS domain, a structural motif frequently associated with coalescence of LLPS proteins. in vivo, we found that removing just 13 amino acids of the LARKS domain abrogated Dan's ability to restrict *hb* relocation to the nuclear lamina and terminate early competence. The competence phenotype was restored upon replacing Dan's LARKS domain with that of other known LLPS proteins or a poly-asparagine tract, known to mediate protein aggregation. Such domain-swapping experiments in the embryo support the conclusion that LARKS-mediated coalescence is essential for Dan's ability to maintain early competence in vivo. Notably, the LARKS domain constitutes a minor fraction of the largely disordered Dan protein, and appears to contribute to only a part of Dan self-association properties. Compared to deleting the DBD alone, co-deletion of the LARKS and DBD domains resulted in smaller Dan hub size but not a total dispersal of Dan protein. At the macroscopic level, distribution of Dan with LARKS deletion is qualitatively similar to full length Dan, though changes to hub formation could be occurring at smaller scales that are not readily visible with whole cell imaging. We do not yet know the specific mechanisms by which Dan regulates genome organization and the role for the LARKS domain in this context. The data suggest that multiple regions of Dan protein contribute to its coalescence. Given that Dan binds widely across the genome, it is possible that Dan tethers multiple genomic sites together through coalescence, and loss of the LARKS domain reduces self-association just enough that a subset of targets, like *hb*, are not efficiently tethered in the nuclear interior. Additionally, it could be that the LARKS domain functions not only to contribute to coalescence but also to regulate interactions with proteins besides Dan, perhaps restricting access of other regulators of nuclear architecture to specific target genes. In such a case, loss of LARKS might not noticeably impact Dan coalescence, but could render genes like *hb* vulnerable to access by factors that relocate them to the nuclear lamina (Fig. 6 bii). The data presented here establish a crucial link between Dan's coalescence properties and in vivo biological function in nuclear architecture and progenitor competence. We envision that Dan might act to stabilize the genome organization associated with the neuroblast's early competence state, and its transient downregulation at mid-embryogenesis allows for new genomic configuration to be adopted. Future experiments that use gain and loss-of-function studies and profile genome-wide chromatin contacts and Dan interacting proteins in neuroblasts will be necessary and fascinating to further understand Dan's role at a mechanistic level. Such studies will provide insights into neural progenitor genome reorganization during development and its role in neural diversification.

## Methods

### Fly lines
Wild-type (*w1118*), *insc-GAL4* (BDSC#8751), *UAS-GFP* (BDSC#32184), *wor-GAL4* (BDSC#56553)[70], *fkh-GAL4* (BDSC#78060)[71], *GAL80ts* (BDSC#7019), *en-GAL4* (chrom II, BDSC#46438), *UAS-hb*[72], *hb*[HA 3]. We[5] generated the following constructs in this paper: *UAS-DanFL*, *UAS-DanGFP*, *UAS-DanMyc* (called DanFL in Fig. 4c), *UAS-DanΔPSQ*, *UAS-PSQ:GFP*, *UAS-DanΔLARKS*, *UAS-DanNNN*, *UAS-DanFUS*, *UAS-DanNup98*. Transgenic flies were generated through PhiC31-mediated insertion at either attP2 or attP40. Flies were raised on a standard cornmeal/molasses medium at 25 °C.

### Plasmids
UAS-DanFL was made by restriction cloning from Dan cDNA LD40883 and UAS plasmid PRVV70 (kind gift from Dr. Richard Mann, Columbia University). PRVV70 has a phi-C31 integrase binding site for genomic integration, a UAS/Gal4 binding site, and a multiple cloning site containing XbaI and KpnI restriction sites. The Dan 5′UTR and coding region were extracted by PCR from LD40883 (#5984 DRGC), and XbaI and KpnI restriction sites were added. Ac5-DanFL was made by DNA assembly using NEBuilder HiFi DNA Assembly Cloning Kit (NEB #E5520S) from pAc5.1C-FLuc-V5His6 (Addgene plasmid #21183), Ac5-STABLE1-neo (Addgene plasmid #32425) and UAS-DanFL. All other plasmids were generated by DNA assembly or site-directed mutagenesis using UAS-DanFL or Ac5-DanFL as the backbone.

For domain-swapping experiments in which the Dan LARKS domain was replaced, the following primers were used: (1) NNN primers, F=caacaacaacaacaacaaccttctgggaccgccggag, R=ttgttgttgttgttgtt gttatccatcttatcgcccaggg. (2) FUS primers, F=cgtggccgtggccggccctcttct gggaccgccggag, R=gccgccacggccgtttccatccatcttatcgcccaggg, (3) Nup98 primers, F=ttcggaacatcccttctgggaccgccggag, R=gttgccgaagcca tccatcttatcgcccaggg.

### S2 cell culture
S2 cells were sourced from ThermoFisher Scientific (#R69007) and were grown in T25 flasks (Thermo Scientific, #156367) at 25 °C in Schneider's *Drosophila* Medium (Gibco #21720024) with 10% heat inactivated FBS, and cultured following standard protocols[73]. For transfection, $1 \times 10^6$ cells were seeded onto a 35 mm glass bottom dish (Cellvis #D35-20-1.5-N). After 24 h, cells were transfected with 500 ng plasmid DNA and a 1:50 Effectene transfection reagent ratio (Qiagen #301425). FRAP measurements were performed 24hrs after transfection and incubation at 25 °C. For condensate size comparisons, constructs were transfected in S2 cells split from the same starting culture and incubated side by side for 36hrs under the same conditions.

### Antibodies
**Primary antibodies.** anti-Eve (1:50, Mouse, #3C10 Developmental Studies Hybridoma Bank), anti-HA (1:500, Rat, #11867423001 Sigma Roche), anti-Dan (1:1000, Rabbit, (Kohwi et al., 2013)), anti-Myc (1:100, Rabbit, #ab9106 Abcam), anti-MSL2 (1:300, Rabbit, kind gift from Dr. Mitzi Kuroda, Harvard University), anti-Dpn (1:100, Rat, #ab195172 Abcam), anti-Wor (1:100, Rat, #ab196362 Abcam), anti-Lamin (1:1000, Rabbit, R-836, kind gift from Dr. Paul Fischer, Stonybrook).

**Secondary antibodies.** Goat anti-Rabbit IgG, Alexa Fluor 555 (#A21429 Invitrogen), Donkey anti-Rat IgG, Biotin-SP (#712065153 Jackson ImmunoResearch), Donkey anti-Mouse IgG, Alexa Fluor 647 (#715605151 Jackson ImmunoResearch), Goat anti-Mouse IgG, DyLight 550 (#A90516D3 Bethyl Laboratories), Goat anti-Rabbit IgG, Alexa Fluor 488 (#A11034 Invitrogen), Donkey anti-Rat IgG, DyLight 550 (#A110337D3 Bethyl Laboratories). All Invitrogen secondaries were used at 1:400, Bethyl lab secondaries at 1:100. Streptavidin Cy3 (1:500, #SA1010 Invitrogen). DAPI (200 ng/ml, #D3571 Invitrogen). NucBlue

(#R37605, ThermoFisher) was used as the DNA stain for live S2 cell imaging.

## Immunostaining

Embryos were immunostained following standard protocols[7,74]. Briefly, embryos were fixed in a 1:1 mixture of 4% formaldehyde in PEM buffer (0.1 M Pipes, 1 mM MgSO4, 2 mM EGTA) and N-heptane and rocked for 22 min at room temperature. Following fixation, embryos were devitellinized by vigorous shaking in a 1:1 mixture of methanol:heptane (methanol cracking) and washed with PBS-0.1% Tween 20 (PBST) before staining. Embryos were incubated in primary antibodies diluted in PBST overnight at 4 °C, secondary antibodies at room temperature for 1.5 hours, and streptavidin for 20 min at room temperature. After final washing, embryos were placed in 90% glycerol before imaging. For DNA FISH experiments, male and female embryos were used; for early competence assay experiments, only male embryos were scored, as the presence of y chromosome marker, MSL2, indicate lack of the repressive Gal80 driver (see Supplementary Fig. 2 for details). Salivary glands were fixed and immunostained following standard protocol with minor modifications[75]. Briefly, salivary glands were dissected from L3 larvae grown at 16 °C after a 4-h heat induction at 29 °C to express Dan. Dissected glands were fixed for 90 s in 2% PFA/45% Acetic Acid, then transferred to a droplet of 45% Acetic acid on a siliconized coverslip. The glands were squashed, flash frozen in liquid nitrogen, then washed in PBST. Squashes were incubated in antibodies diluted in PBST overnight at 4 °C, secondary antibodies at room temperature for 1 h, and DAPI for 2 mins at room temperature.

## Photobleaching assays

**Fluorescence recovery after photobleaching (FRAP).** Cells were imaged every 500 ms for 60 frames on a Zeiss LSM 700 Confocal with a 63x objective with Immersol 518 F immersion oil (Carl Zeiss #444960). After the third image, a circular region with a diameter ~1 μm was bleached using the 488 nm laser at 100% power for 54.03μs, after which imaging continued for the remaining frames. Recovery was measured as fluorescence intensity of the photobleached or control area normalized to the intensity of the bleached spot immediately after bleaching.

**Fluorescence loss in photobleaching (FLIP).** A z-stack of a transfected cell was taken prior to bleaching, after which the cell was imaged every 500 ms for 10 frames, and bleached using the same conditions as for FRAP. After the 10 frames, 3D imaging of the entire nucleus was repeated. 3D imaging was performed a total of 5 times, and bleaching a total of 4 times. Bleaching was repeated on the same condensate or in the same spot outside the nucleus. Fluorescence loss was measured in either the whole cell, by measuring the relative fluorescence intensity of combined z-stack over time, or in individual condensates by measuring the relative fluorescence intensity of individual condensates in a single cell over time. Recovery of the bleached spot immediately before and after bleaching was measured as fluorescence intensity of the photobleached or control spots normalized to the intensity of the bleached spot immediately after bleaching.

## Immuno-DNA FISH

DNA in situ hybridization (DNA-FISH) was performed as previously described[3,7] as follows. Generation of *hb* DNA FISH probe: probes were generated by amplifying genomic DNA spanning a ~ 10 kb region at the *hb* locus using PCR (see Table S1 for primer sequences). Fluorescent probes were made from the PCR products using the DNA FISH Tag kit (Thermo Fisher Scientific, #F32948). Probe hybridization: Rehydrated embryos were treated with RNaseA (150 mg/ml) for 2 h at room temperature and washed 1 h in PB-0.3%Triton-X (PBTx). Embryos were gradually stepped into 100% pre-hybridization mixture (pHM: 50% formamide; 4XSSC; 100 mM NaH2PO4, pH 7.0; 0.1% Tween 20) by a series of PBTx:pHM washes for 20 min each. Embryos were prehybridized 1 h at 37°C, denatured 15 min at 80°C. DNA probe was diluted to 2 ng/μl in hybridization buffer (10% Dextran sulfate, 50% deionized formamide, 2XSSC, 0.5 mg/ml Salmon Sperm DNA) and denatured for 5 min at 95°C. The hybridization probe mixture was added to the embryos, and embryos were rocked for 16 h at 37 °C after which they were washed in a series of decreasing formamide/0.3% CHAPS solutions at 37°C with final washes at room temperature. Embryos were subsequently immunostained according to methods described above, mounted in vectashield (Vectorlabs) and imaged on a confocal microscope, Zeiss LSM 700.

## Imaging and analyses

Images of immunostained embryos and salivary glands as well as FRAP and FLIP assays were conducted using the Zen software on a Zeiss LSM 700 Confocal microscope. Imaging of live S2 cells and dissociated embryonic cells was done using the NIS-Elements software on a W1-Yokogawa Spinning Disk Confocal. For 3D imaging for DNA FISH, images were taken at a 0.4 μm step size, and pinholes were adjusted to have equal optical section thickness for all channels. The z-plane in which the FISH signal was strongest was identified and the shortest distance to the nuclear lamina was measured. FISH signals were scored as on-lamin if the FISH signals and the lamin signals pixel-overlapped; FISH signals that were not pixel-overlapping, but where within 0.4 μm of the lamin signal were scored as near-lamin. All image analyses were done using FIJI[76].

## S2 cell related analyses

**Condensate analyses.** Condensates were counted manually for each nucleus. To analyze shape, each condensate was converted to grayscale and threshold segmentation was performed in FIJI. Minor adjustments to the threshold were made to minimize background that could affect condensate shape. Shape was measured using the FIJI built-in circularity calculations based on the formula circularity = $4p$ (area/perimeter$^2$). Condensate fusion: A kymograph representing the fusion of two condensates was made by drawing a pixel-thin ROI with a consistent length through the centers of the two condensates of interest in every frame. The ROIs were then placed side by side in chronological order. Protein expression levels: Mean nuclear GFP was calculated by summing the z-stack, manually outlining the nucleus as an ROI, and measuring mean gray value of the nuclear ROI using FIJI.

## Other analyses

**Statistics.** We applied standard *t* tests for normally distributed data, and Mann–Whitney test for nonparametric data using Prism v9. All statistical tests are two-sided. Kolmogorov-Smirnov test of normality was used to determine whether *t* test or Mann–Whitney test will be used. Statistical significance was classified as follows: * < 0.05, ** < 0.01, *** < 0.001, **** < 0.0001. Graphs show each data point used for statistical analysis, and bars indicate mean and standard deviation for *t* tests, and median and interquartile range for Mann–Whitney tests. LARKS propensity profile link: https://srv.mbi.ucla.edu/LARKSdb/Login/index.php.

**Protein structure prediction.** Dan intrinsic disorder was predicted using IUPred3, using IUPred3 long disorder analysis and medium smoothing[37].

**Competence assay.** Neurons expressing Eve and HA were identified based on the presence of Eve and HA signal above background. NB7-1 lineage Eve+ neurons can be identified by their medial localization within the ventral nerve cord and easily distinguished from the deeper RP2, aCC/pCC neurons from other lineages that are also Eve+ as well as the Eve lateral cells that form a cluster of Eve+ neurons lateral to the NB7-1 progeny. NB7-1 early competence has been described

previously[1,3,4,7], and quantification of control genotypes were consistent across studies and observers[3,7,19], indicating the reproducibility and robustness of the assay.

### Reporting summary

Further information on research design is available in the Nature Portfolio Reporting Summary linked to this article.

## Data availability

The datasets generated during the current study are available from the corresponding author upon request. All requests for information on reagents, experimental protocols, and other inquiries should be made to the corresponding author, Minoree Kohwi. Source data are provided with this paper.

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

## Acknowledgements

We thank members of the Kohwi Lab for scientific discussion and feedback. We thank Ms. Sofiya Patra and Dr. Tanguy Lucas for technical assistance with experiments. We also thank Drs. Daniel Kalderon, Stavros Lomvardas, Richard Mann, Wes Grueber, Carol Mason for scientific feedback.

## Author contributions

G.B. and M.K. designed and executed the experiments, analyzed data and wrote the manuscript. R.J.C. assisted with and quantified the cell culture and embryo staining experiments. M.J.L. generated constructs and performed the cell culture assays. N.M. performed the DNA FISH experiments. M.K. received support from NIH grants R00HD072035 and

R01HD092381, Rita Allen Foundation, Whitehall Foundation, and Hirschl Trust Research Scientist Program. G.B. received support from the NIH T32 training grant GM879818.

## Competing interests

The authors declare no competing interests.

## Additional information

**Peer review information** : *Nature Communications* thanks the anonymous reviewers for their contribution to the peer review of this work. A peer review file is available.

