## [Peer Review File · Nature Communications]

Dan coalesces in the nucleus and regulates nuclear architecture and progenitor competence *in vivo*Editorial Note: This manuscript has been previously reviewed at another journal. This document only contains reviewer comments and rebuttal letters for versions considered at Nature Communications.

REVIEWER COMMENTS

Reviewer #1 (Remarks to the Author):

The authors have made several changes to this study and improved the manuscript. However, several of my original concerns remain. Referring to my original points:

1 - 6: OK.

7: Fig 2: Briefly, I wanted the authors to stain S2 cells expressing un-fused Dan with anti-Dan antibodies, to confirm that the nuclear sub-localization of Dan in tissue culture is not an artefact stemming from the exclusive use of a Dan-GFP fusion protein. They now argue that studies show that fixation affects LLPS protein distribution and have chosen not to add this experiment. However, arguing that a straightforward experiment will likely fail is not logical; what does antibody staining for Dan in S2 cells look like? My concern remains.

8: Fig 3b: What is the behaviour of a Dan-delta-LARKS construct in the cell culture assay? They found that Dan-delta-LARKS distributed similarly to full-length Dan. This in part goes against their model, because Dan-delta-LARKS is unable to alter temporal progression in the NB3-1 lineage but still distributes similarly to full-length Dan. In this re-submission they argue that the LARKS domain only partly contributes to LLPS, and that other parts of the Dan protein contribute to LLPS behaviour of the Dan protein. But what are these other parts of Dan then? My concern remains.

9 - 13: OK.

Reviewer #4 (Remarks to the Author):

In this manuscript, the authors employ a cell-fate decision model (neuroblast competency) to probe how genome organization and condensate formation may play a functional role in this system. They identify that Dan, a transiently expressed protein with DNA-binding and self-associating domains, forms condensates with a diffuse distribution. While normally Dan expression is repressed over development (and correlates with loss of early competency), they show that over-expression provides an extension of this competency and loss of nuclear reorganization. Interestingly, through multiple assays (fixed cells in an embryo, live imaging in a embryo-derived line) that both the DNA binding and the LARKS domains are important to maintain this competency. Interestingly, while loss of PSQ completely alters condensate organization in the nucleus (from diffuse, DNA colocalized to a handful at the lamina), the loss of LARKS affects competency without such a large-scale reorganization. These data provide a partially complete, yet interesting pieces of a puzzle, of how condensate properties (not just formation but their organization and interaction with other molecules) may contribute to function. The replacement LARK mutant experiments are compelling.

One concern: If the LARKS don't abrogate condensate distribution (broadly), how then do the authors imagine does the effect still lead in loss of competency. The authors should discuss whether the effects may stem from loss of loci-specific interaction (i.e., distributions look broadly similar but there is a hb gene-LARKS domain specificity) or whether there are condensate-independent mechanisms that tie LARK loss to loss of function. While I am hesitant to recommend more experiments, even 1 or 2 along these lines could dramatically add clarity. I don't believe it is necessary to add this if sufficiently discussed otherwise.

KEY: All reviewer comments are in **black text**.
Our responses are in **blue text**.
Section describing how we modified Figures or Texts are in **red text**.

We would like to express our gratitude to the reviewers for the comments and feedback on our manuscript. Please find below our responses to the comments.

Point by point response to Reviewer #1

The authors have made several changes to this study and improved the manuscript. However, several of my original concerns remain. Referring to my original points:

7: Fig 2: Briefly, I wanted the authors to stain S2 cells expressing un-fused Dan with anti-Dan antibodies, to confirm that the nuclear sub-localization of Dan in tissue culture is not an artefact stemming from the exclusive use of a Dan-GFP fusion protein. They now argue that studies show that fixation affects LLPS protein distribution and have chosen not to add this experiment. However, arguing that a straightforward experiment will likely fail is not logical; what does antibody staining for Dan in S2 cells look like? My concern remains.

We thank the reviewer for this feedback. We have performed additional S2 cell culture experiments expressing Dan^{FL} (FL, full length) protein with and without the GFP tag to test whether the GFP tag has any effects on Dan protein distribution in S2 cells (see figure below). We found that both Dan and Dan:GFP protein distributed similarly in nuclei, with punctate foci among a more uniform distribution. This pattern is consistent with Dan staining in neuroblasts of fixed embryos as well as live S2 cells expressing Dan^{FL}:GFP. We also note that both GFP-tagged and non-tagged Dan extensively overlaps with chromatin (DAPI stain) throughout the nucleus but avoids DAPI dense, heterochromatin sites, such as the pericentromeric heterochromatin. This is consistent with Dan distribution in polytene chromosomes. The experiment comparing tagged and non-tagged Dan protein was performed on fixed and stained S2 cells to visualize Dan protein. We also present the following experiments that support that the GFP tag is not the driver of Dan protein distribution.

(1) In live S2 cells, GFP alone distributes uniformly, without puncta seen in Dan^{FL}:GFP.

(2) In live S2 cells, Dan^{FL}:GFP co-localizes with chromatin and forms discrete puncta, whereas Dan^{ΔPSQ}:GFP, which lacks the DNA-binding domain, forms large hubs that avoid chromatin. PSQ:GFP, which removes all of the disordered domains while keeping the DNA-binding domain intact, distributes uniformly in the nucleus. The GFP tag is common between all three constructs, and thus the difference in Dan protein distribution between them can be attributed to the structural changes in Dan.

(3) Finally, misexpression of Dan:GFP in neuroblasts in the embryo has the same effects on early competence as Dan with no GFP tag (P value not significant), indicating that the GFP is functionally inert on Dan function *in vivo*. This is in Fig. 2aiii.

Together, we believe that the GFP is not the driver of Dan protein distribution nor does it impact Dan function *in vivo*.

Figure legend: Dan with and without GFP tag distributes similarly in S2 cells. **A)** z-projection of a group of representative S2 cells, some of which express the transfected Dan construct, are shown. Dan^{FL}:GFP is on the left, and Dan^{FL} alone is on the right. Both sets of cells were fixed 48hrs after transfection and stained with anti-Dan antibody, anti-GFP antibody, and DAPI to confirm that only Dan^{FL}:GFP has GFP signal. **B)** single z-plane of several examples of S2 cells expressing Dan^{FL}:GFP (left side) or Dan alone (right right) are shown as separate channels of Dan and DAPI. In both sets, Dan protein distributes as puncta of various sizes among a more uniform distribution and overlaps extensively with chromatin (DAPI).

8: Fig 3b: What is the behaviour of a Dan-delta-LARKS construct in the cell culture assay? They found that Dan-delta-LARKS distributed similarly to full-length Dan. This in part goes against their model, because Dan-delta-LARKS is unable to alter temporal progression in the NB3-1 lineage but still distributes similarly to full-length Dan. In this re-submission they argue that the LARKS domain only partly contributes to LLPS, and that other parts of the Dan protein contribute to LLPS behaviour of the Dan protein. But what are these other parts of Dan then? My concern remains.

We thank the reviewer for this question and appreciate the opportunity to include further discussion on the role that LARKS plays in Dan function on the neuroblast genome. The Dan^{ΔLARKS} distribution (Fig. 4a) does not go against our model. We show that the LARKS domain confers self-association properties to Dan, and that this property is essential for competence regulation *in vivo*, independent of the DNA-binding property. There are several points to consider here.

(1) The LARKS domain does contribute to Dan coalescence, because the Dan^{ΔPSQ} condensate size decreases upon co-deletion of the LARKS domain. However, the LARKS domain is only a small region (ΔLARKS removes only 13 amino acids) of an otherwise mostly disordered protein. Thus, deleting the LARKS domain alone does not remove Dan's ability to coalesce entirely, but rather it may be reducing efficiency for coalescence.

(2) While the distribution of Dan^{ΔLARKS} is similar to Dan^{FL}, this is a qualitative assessment at the macroscopic level. We do not know whether LARKS deletion affects Dan condensate size at the genomic level, something that we cannot readily visualize with live imaging of whole cells. However, we have tested LARKS function *in vivo*, and we show that LARKS deletion results in loss of Dan's ability to maintain the *hb* gene in the nuclear interior and loss of ability to maintain early competence. Importantly, we show that replacing the LARKS domain with that from other known phase-separating proteins restores this functionality in the embryo.

(3) LARKS deletion does not impact DNA-binding. When the PSQ domain is deleted, Dan can no longer bind chromatin, but not when the LARKS domain is deleted (polytene chromosome staining experiment, Fig. 4bi). This shows that Dan^{ΔLARKS} loss of ability to extend early competence is not due to an indirect effect of reduced DNA-binding.

(4) The PSQ domain alone, on the other hand, effectively removes all the disordered regions, and this results in uniform distribution of the protein, supporting that LARKS contributes to, but is not solely responsible for, Dan coalescence (in the embryo, the PSQ domain alone is not able to extend competence, Supp. Fig. 2b).

Taken together, the data indicate that Dan protein distribution is driven by its self-association and its DNA-binding properties. While LARKS may contribute only partially to Dan's total coalescence, LARKS-mediated coalescence properties underlie Dan's function in neuroblast genome organization and competence regulation. How mechanistically Dan's coalescence properties impacts genome organization at the chromatin level, and the specific role that LARKS plays, is yet unknown. Addressing these questions will require technically challenging experiments of chromatin contact profiling and identification of Dan interacting protein partners specifically in neuroblasts obtained from embryos expressing various Dan structural mutants. While beyond the scope of the current work, these are important future studies. We have modified the Discussion section to include possible mechanisms for how LARKS contributes to neuroblast nuclear architecture and discuss future experimental directions.

Our changes to the text:

-We have presented the above points describing the LARKS data with more clarity in the section "A LARKS domain is essential for Dan's ability to regulate competence *in vivo*," lines 250-277.

-We have also provided additional text to discuss LARKS effects on Dan coalescence versus its effects on early competence, discussing possible mechanisms and future experiments to address them. These are in lines 382-413.

-Finally, we have also modified Fig 6 (model) and legends to include these discussion points.

"While LARKS deletion partially reduces Dan coalescence, it abrogates Dan's ability to retain the *hb* gene in the nuclear and maintain the early competence state. Perhaps this occurs through reduced efficiency to tether some target genes to Dan hubs in the nuclear interior and/or restrict their interaction with other gene regulators. Future studies are required to determine mechanisms underlying Dan regulation of genome organization."

Point by point response to Reviewer #4

In this manuscript, the authors employ a cell-fate decision model (neuroblast competency) to probe how genome organization and condensate formation may play a functional role in this system. They identify that Dan, a transiently expressed protein with DNA-binding and self-associating domains, forms condensates with a diffuse distribution. While normally Dan expression is repressed over development (and correlates with loss of early competency), they show that over-expression provides an extension of this competency and loss of nuclear reorganization. Interestingly, through multiple assays (fixed cells in an embryo, live imaging in a embryo-derived line) that both the DNA binding and the LARKS domains are important to maintain this competency. Interestingly, while loss of PSQ completely alters condensate organization in the nucleus (from diffuse, DNA colocalized to a handful at the lamina), the loss of LARKS affects competency without such a large-scale reorganization. These data provide a partially complete, yet interesting pieces of a puzzle, of how condensate properties (not just formation but their organization and interaction with other molecules) may contribute to function. The replacement LARK mutant experiments are compelling.

One concern: If the LARKS don't abrogate condensate distribution (broadly), how then do the authors imagine does the effect still lead in loss of competency. The authors should discuss whether the effects may stem from loss of loci-specific interaction (i.e., distributions look broadly similar but there is a hb gene-LARKS domain specificity) or whether there are condensate-independent mechanisms that tie LARK loss to loss of function. While I am hesitant to recommend more experiments, even 1 or 2 along these lines could dramatically add clarity. I don't believe it is necessary to add this if sufficiently discussed otherwise.

We thank the reviewer for the feedback and thoughtful questions. It is indeed fascinating that the LARKS domain has a large effect on *hb* gene positioning despite contributing only a minor effect on Dan protein coalescence. We do not yet know how mechanistically Dan's coalescence properties impacts genome organization globally, and whether LARKS-mediated coalescence impacts a subset of genes and/or impacts the recruitment of a subset of regulators to specific target sites. These questions will require profiling neuroblast-specific chromatin contacts and Dan protein interactions using embryos harboring various Dan structural mutants. Though beyond the scope of the current work, such efforts will be important to understanding how genome organization is regulated during development and how it impacts downstream cellular events, such as a cell fate specification. We have modified the Discussion section to include possible mechanisms for how LARKS contributes to neuroblast nuclear architecture and discuss future experimental directions.

Our changes to the text:

-We have modified the text to present the the LARKS data with more clarity in the section "A LARKS domain is essential for Dan's ability to regulate competence *in vivo*," lines 250-277.

-We have provided additional text to discuss LARKS effects on Dan coalescence versus its effects on early competence, discussing possible mechanisms and future experiments to address them. These are in lines 382-413.

-Finally, we have also modified Fig 6 (model) and legends to include these discussion points.

"While LARKS deletion partially reduces Dan coalescence, it abrogates Dan's ability to retain the *hb* gene in the nuclear and maintain the early competence state. Perhaps this occurs through reduced efficiency to tether some target genes to Dan hubs in the nuclear interior and/or restrict their interaction with other gene regulators. Future studies are required to determine mechanisms underlying Dan regulation of genome organization."

REVIEWERS' COMMENTS

Reviewer #1 (Remarks to the Author):

I had two remaining concerns, which the authors now have attempted to address.

7: Fig 2: Briefly, I wanted the authors to stain S2 cells expressing un-fused Dan with anti-Dan antibodies, to confirm that the nuclear sub-localization of Dan in tissue culture is not an artefact stemming from the exclusive use of a Dan-GFP fusion protein. They have conducted this experiment now and show images pointing to that Dan and Dan-GFP both display nuclear sub-localisation in S2 cells. This is re-assuring.

8: Fig 3b: What is the behaviour of a Dan-delta-LARKS construct in the cell culture assay? They found that Dan-delta-LARKS distributed similarly to full-length Dan. This in part goes against their model, because Dan-delta-LARKS is unable to alter temporal progression in the NB3-1 lineage but still distributes similarly to full-length Dan. In this re-re-submission they argue again that the LARKS domain only partly contributes to LLPS. However, the data supporting this are not convincing. The nuclear staining of Dan[FL]-GFP and Dan[LARKS]-GFP looks very similar (Fig4 ai) and the quantification of droplet size (Fig 4bii) is not convincing. Regarding the latter, they should depict the individual data points in the graph and state in the figure legend the statistical analysis conducted.

REVIEWERS' COMMENTS

Reviewer #1 (Remarks to the Author):

I had two remaining concerns, which the authors now have attempted to address.

7: Fig 2: Briefly, I wanted the authors to stain S2 cells expressing un-fused Dan with anti-Dan antibodies, to confirm that the nuclear sub-localization of Dan in tissue culture is not an artefact stemming from the exclusive use of a Dan-GFP fusion protein. They have conducted this experiment now and show images pointing to that Dan and Dan-GFP both display nuclear sub-localisation in S2 cells. This is re-assuring.

We appreciate the feedback.

8: Fig 3b: What is the behaviour of a Dan-delta-LARKS construct in the cell culture assay? They found that Dan-delta-LARKS distributed similarly to full-length Dan. This in part goes against their model, because Dan-delta-LARKS is unable to alter temporal progression in the NB3-1 lineage but still distributes similarly to full-length Dan. In this re-re-submission they argue again that the LARKS domain only partly contributes to LLPS. However, the data supporting this are not convincing. The nuclear staining of Dan[FL]-GFP and Dan[LARKS]-GFP looks very similar (Fig4 ai) and the quantification of droplet size (Fig 4bii) is not convincing. Regarding the latter, they should depict the individual data points in the graph and state in the figure legend the statistical analysis conducted.

Thank you for this feedback. We agree and put the individual data points in the graph with the error bars and statistics.